# Aquatic Productivity under Multiple Stressors

**Donat-P. Häder** [1],* and **Kunshan Gao** [2,3]

1 Department of Biology, Friedrich-Alexander University Erlangen-Nürnberg, Neue Str. 9, 91096 Möhrendorf, Germany
2 State Key Laboratory of Marine Environmental Science, College of Ocean and Earth Sciences, Xiamen University, Xiamen 361102, China
3 Co-Innovation Center of Jiangsu Marine Bio-industry Technology, Jiangsu Ocean University, Lianyungang 222000, China
* Correspondence: donat@dphaeder.de

**Abstract:** Aquatic ecosystems are responsible for about 50% of global productivity. They mitigate climate change by taking up a substantial fraction of anthropogenically emitted $CO_2$ and sink part of it into the deep ocean. Productivity is controlled by a number of environmental factors, such as water temperature, ocean acidification, nutrient availability, deoxygenation and exposure to solar UV radiation. Recent studies have revealed that these factors may interact to yield additive, synergistic or antagonistic effects. While ocean warming and deoxygenation are supposed to affect mitochondrial respiration oppositely, they can act synergistically to influence the migration of plankton and $N_2$-fixation of diazotrophs. Ocean acidification, along with elevated $pCO_2$, exhibits controversial effects on marine primary producers, resulting in negative impacts under high light and limited availability of nutrients. However, the acidic stress has been shown to exacerbate viral attacks on microalgae and to act synergistically with UV radiation to reduce the calcification of algal calcifiers. Elevated $pCO_2$ in surface oceans is known to downregulate the CCMs ($CO_2$ concentrating mechanisms) of phytoplankton, but deoxygenation is proposed to enhance CCMs by suppressing photorespiration. While most of the studies on climate-change drivers have been carried out under controlled conditions, field observations over long periods of time have been scarce. Mechanistic responses of phytoplankton to multiple drivers have been little documented due to the logistic difficulties to manipulate numerous replications for different treatments representative of the drivers. Nevertheless, future studies are expected to explore responses and involved mechanisms to multiple drivers in different regions, considering that regional chemical and physical environmental forcings modulate the effects of ocean global climate changes.

**Keywords:** aquatic ecosystems; global climate change; ocean acidification; deoxygenation; solar UV radiation

## 1. Introduction

The marine ecosystems cover 70.8% of our planet. Their primary productivity rivals that of all terrestrial ecosystems taken together [1], even though their standing crop is only about 1% of their counterparts on land [2,3]. The primary producers in these ecosystems include macroalgae, which are mainly confined to coastal habitats because they are sessile [4], with a few exceptions, such as members of the genus *Sargassum* which are found floating in the open ocean [5]. The highest concentration of marine biomass is found at higher latitudes and near the coasts. The majority of the aquatic primary producers consists of prokaryotic and eukaryotic phytoplankton both in freshwater and marine ecosystems [6,7]. It is interesting to note that a large share of the phytoplankton long escaped the identification and quantification due to their minute sizes (0.2–2 μm) [8]. Only recently, modern technology, including tyramide signal amplification/fluorescence [9], flow cytometry [10] and molecular genetics [11], showed that picoplankton constitutes a major share of the

marine phytoplankton, with *Prochlorococcus* and *Synechococcus* being the dominant contributors [12]. Eukaryotic phytoplankton includes diatoms, chlorophytes, dinoflagellates and chrysophytes [13]. The highest biomass densities of primary producers are found in coastal regions that are sustained by high terrestrial nutrient runoffs, as shown by remote sensing from satellites [14] (Figure 1).

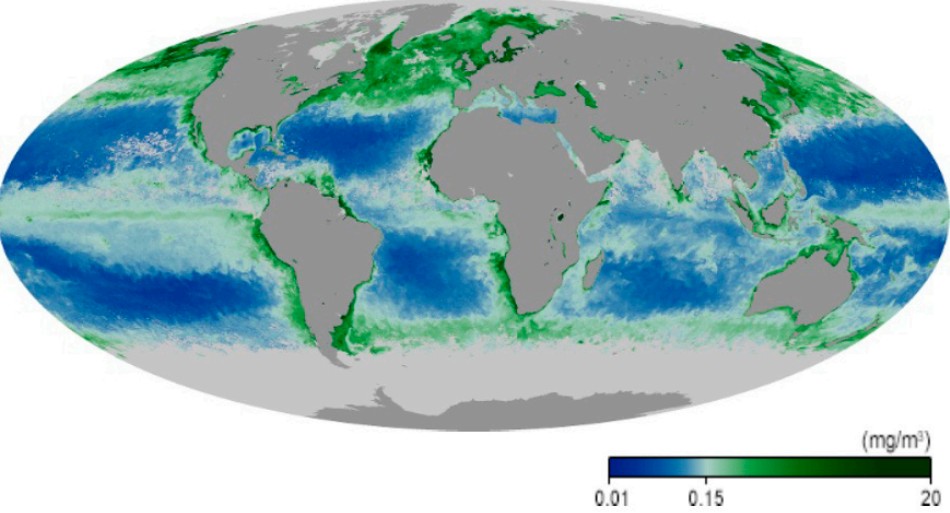

**Figure 1.** Global chlorophyll *a* concentration, as seen in a satellite image from May 2022. Courtesy of https://earthobservatory.nasa.gov/global-maps/MY1DMM_CHLORA/MYD28M (accessed on 1 February 2023).

Because of the requirement for solar radiation, the primary aquatic producers are restricted to the photic zone, the lower limit of which is defined as the depth where the light level has decreased to 1% of the surface irradiance [15]. This is the light level at which respiration compensates photosynthetic oxygen production. The physical depth of the photic zone depends on the concentrations of organic and inorganic dissolved and particulate matters, which are much higher in coastal than in oligotrophic open oceanic waters [16]. The prokaryotic and eukaryotic organisms form the basis of many extended food webs and sustain zooplankton, invertebrates, fish and mammals and provide food for the growing human population outcompeting the production of meat from terrestrial animals in many regions of the world [17].

The oceans absorb about 50–60 Petagram (PG) of anthropogenically released carbon per year. The biosphere in the oceans and on land absorbs about 45% of the anthropogenically released carbon dioxide [18]. The $CO_2$ concentration has increased from about 270 ppm before the industrial revolution to about 420 ppm today [19]. During the period from 2009 to 2018 the oceanic sink for anthropogenic carbon was about 2.5 ± 0.6 PG C per year, which drastically reduces the effects of global warming [18]. Part of the absorbed atmospheric $CO_2$ is taken up by phytoplankton in the top layer (photic zone) of the water column and sediments to the deep sea when the organisms die or in the form of fecal pellets as marine snow, a process called marine biological $CO_2$ pump [20,21]. About 1% of the organic biomass is thought to sink to the pelagic bottom during about 10 years over a distance of 4000 m on average, where it stays for millions of years [22].

The productivity of aquatic organisms depends on a plethora of environmental factors and is affected by a multitude of stressors. During the end of the last century, scientists concentrated on studying the effects of individual factors in laboratory experiments, which copy the natural situation only inadequately. This was a valid approach to understand the biochemical and metabolic mechanisms in the cells. Later on, these studies were carried out in natural waters, often using mesocosms, enclosed volumes of natural water [23,24]. However, these factors do not operate individually but interact in a synergistic, antagonistic

or additive way. This complicates the research, which has to be carried out under natural and changing conditions to provide ecologically relevant results.

## 2. Global Climate Change

Increasing anthropogenic emissions of greenhouse gases result in rising temperatures in the Earth's atmosphere, though the oceans have absorbed more than 90% of the Earth's back heating. Since 1979, the mean global air temperature has increased by 0.27 °C per decade [25], and the latest (6th) IPCC report predicts that limiting global warming to 1.5 °C will require drastic measures [26]. In contrast, the sea surface temperature increased at about 0.13 °C per decade due to the large buffering capacity of the oceans [27] (Figure 2). However, the increment is far from being uniform on the planet [28]. This is especially evident in the Arctic, where temperatures rise much faster than in most other parts of the world. This is in part due to a feedback mechanism: Ice and snow scatter and reflect the incoming solar radiation to a high degree [29]. As the ice melts, the open soil and water absorb solar radiation more strongly, heating the land and sea. As a consequence, the summer ice extent in the Arctic Ocean has decreased by about 45% during the last three decades [30,31]. This has dramatic consequences for the water column. The ice and snow cover protected the underlying photic zone from impinging solar UV-B radiation [32]. In contrast, the higher temperatures and increased impact of visible radiation supports a fast growth of phytoplankton in the upper layer which fosters an increase in consumer biomass [33]. Rapid ice melting reduces the salinity of the water and increases the amount of dissolved and particulate matter [34], which affects growth and species composition of the phytoplankton communities.

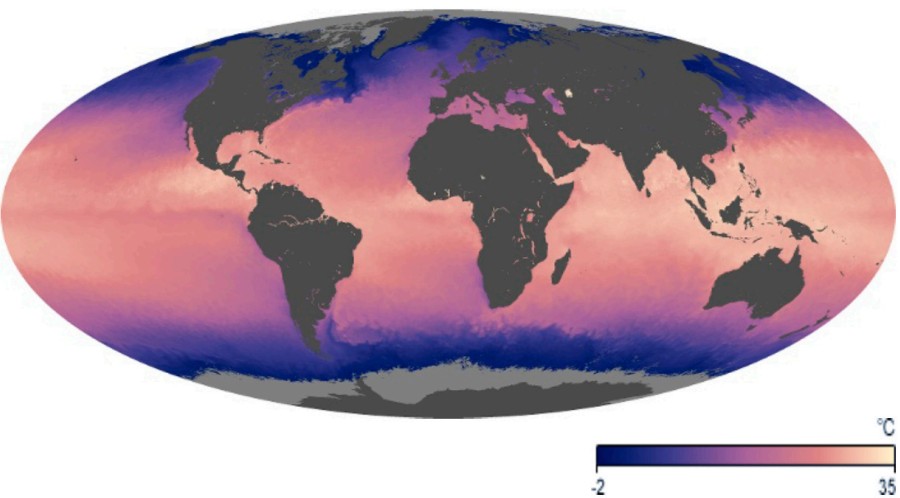

**Figure 2.** Global water surface temperatures as seen in a satellite image from May 2022. Courtesy of https://earthobservatory.nasa.gov/global-maps/MY1DMM_CHLORA/MYD28M (accessed on 1 February 2023).

All organisms have specific thermal windows concerning their tolerance of thermal stress. We can define a lower limit, an optimum temperature and an upper limit. Tropical and subtropical corals have been found to die due to extensive heat that exceeds their permissive upper temperature limit [35,36]. Extended exposure to elevated temperatures results in the expulsion of their symbiotic zooxanthellae, which are photosynthetic dinoflagellates [37,38], resulting in massive bleaching and causes starvation [39,40]. The first signs of thermally induced damage can be detected by pulse-amplitude-modulated (PAM) chlorophyll fluorescence and photorespirometry. Exposure of *Stylophora pistillata* to 34 °C for 4 h resulted in a strong non-photochemical quenching, which indicates that the absorbed solar energy is no longer available to drive photosynthesis but is dissipated as heat [39]. Furthermore, the photosynthetic oxygen production and the quantum yield were

drastically reduced. Bleaching is further aggravated by exposure to solar UV radiation especially at lower depths [41], as found in the sensitive *Pocillopora meandrina* up to a depth of 20 m, even though corals utilize a UV-absorbing pigment to protect them from UV radiation [42].

The first modern beaching events (2014–2017) affected more than two-thirds of all coral reefs in the ocean [43], such as the Great Barrier Reef [44], and were further aggravated by a strong El Niño phenomenon [45], resulting in a 2 °C increase to >32 °C [46]. Elevated temperature and higher ammonium concentrations also delayed or stopped larval development, as shown for *Diploria strigosa* [47].

The temperature in the Mediterranean Sea is rising three times faster than in the global oceans [48]. Typical seagrasses and macroalgae in the area, such as *Posidonia oceanica*, *Cystoseira compressa*, *Padina pavonica*, *Caulerpa prolifera* and *Halimeda tuna*, differ in their thermal optima, and their upper lethal limits were found between 28.9 and >34 °C. The highest temperature optimum in this study was detected in *Cymodocea nodosa*. These results indicate that some species will profit from climate-change-induced higher temperatures by outcompeting other species, though little has been documented on their juvenile or spore/gamete stages. Increasing temperatures also change the species composition in phytoplankton assemblages [49], as indicated by comparing several thousand foraminifera communities from pre-industrial times with modern ones. Some species have been found to show a fast adaptation to increasing temperatures. Four diatom species isolated from the tropical Red Sea adapted to 30 °C after 200–600 generations and showed increased optimal growth temperature and their upper tolerated temperature limit [50]. Several toxic phytoplankton species have been found to tolerate increasing oceanic and freshwater temperatures, resulting in harmful algal blooms [51,52], including cyanobacteria such as *Microcystis* [53], diatoms and dinoflagellates [54].

Another option to deal with increasing temperature is a poleward migration [55]. In sessile organisms such as macroalgae and corals, it is a multigenerational process, while in motile forms, this can be achieved on shorter timescales [56]. For example, tropical and subtropical radiolaria have been found to move poleward, and this was especially pronounced during El Niño events [57,58]. The same effect was found in fish, such as cod, which migrate poleward to avoid heat stress and follow their food [59].

Increasing temperatures decrease the solubility of oxygen in seawater, and the $O_2$ concentration has been found to decline in coastal and open ocean habitats over the past five decades [60]. In addition, in coastal regions, deoxygenation is augmented by eutrophication (especially N and P), which results in higher $O_2$ consumption and thus the development of hypoxic "dead zones" [61,62].

Increasing water temperatures result in higher stratification both in the ocean and in freshwater ecosystems [63], resulting in a defined upper mixed layer (UML) in which warmer and thus lighter water is vertically moved by winds and waves. The UML is typically 20 to 100 m deep in the oceans but much shallower in lakes on the order of a few to tens of meters, being shallower during summer seasons. The lower boundary of the UML is the thermocline, which every diver knows when he/she leaves the warm layer and dives down into distinctly colder waters [64]. The thermocline also limits the transport of nutrients-rich deep water up to the UML [65]. Most of the primary producers dwell in the UML, where they receive sufficient amounts of photosynthetically active radiation [66]. Initially, it was reported that increasing temperatures always reduce the depth of the UML (shoaling), thus decreasing the path length of solar radiation and thus exposes the organisms dwelling in this layer to higher UV radiation [67]. More recent studies showed that climate change may have heterogeneous effects [68]. In some cases, higher temperatures indeed result in shoaling of the UML; in others, this effect is counterbalanced or overcome by stronger winds [69]. A long-term international program (1970–2018) using autonomously-diving ocean sensors spread over most of the global oceans has confirmed that, on average, the UML depth in the oceans has decreased by 2.9% per decade, and this corresponds to 5–10 m per decade [70]. In the Southern Ocean, this trend is more

pronounced than in the North Atlantic, while a decreasing mixed layer depth has been found at high Arctic latitudes.

### 3. Ocean Acidification and its Effects

As indicated above, the atmospheric $CO_2$ concentration has increased to above 420 ppmv and is expected to further increase due to anthropogenic emissions from fossil-fuel burning, tropical deforestation and changes in land use [71,72]. An IPCC model (A1F1 based on a business-as-usual emission scenario) predicts an increase of up to 800–1000 ppmv by the year 2100 [73]. Along with the increasing $CO_2$ concentration in the atmosphere, the oceans quickly take up the anthropogenically released $CO_2$, with an approximate rate of 1 million tons per h. Such dissolution of $CO_2$ has been detected down to about 1000 m in depth. Since the oceans have already absorbed more than 30% of the anthropogenically released $CO_2$ [8,74], the marine carbonate chemistry has been being altered, with increased $H^+$ and decreased $CO_3^{2-}$ ions, along with increased concentrations of bicarbonate and total dissolved inorganic carbon. It is known that the oceans have been acidified by over 30% since the industrial revolution, even though the seawater has a high buffering capacity [75,76]. Predictions for the year 2100 indicate an increase by 100–150% of $H^+$, corresponding to a pH drop by 0.3–0.4 units [73,77,78]. The rate of this alteration in the world oceans chemistry is unparalleled in the recent ca. 1 million years [79].

The availability of $CO_2$ is a bottleneck for photosynthesis. Thus, it had been assumed that rising $CO_2$ concentrations in the water could fertilize the oceans by augmenting photosynthetic productivity in phytoplankton and macroalgae [80]. This has also been found for several phytoplankton groups at moderate increases in carbon dioxide [81]. However, many photosynthetic organisms possess $CO_2$-concentrationg mechanisms (CCMs), allowing them to use the large $HCO_3^-$ pool in seawater [82]. For these primary producers, increased concentrations of carbon dioxide may not have a significant effect [82], while for those which lack the CCMs, it appears to be an advantage [83]. Even organisms with CCMs have an advantage of higher $CO_2$ concentrations since they can downregulate this energy-consuming process and can better thrive in low-light conditions. For example, in diatoms, photosynthetic productivity and growths are enhanced at elevated $CO_2$ concentrations and low light levels, but they are impaired at higher irradiances due to the compounded impact of acidification and light stress [72]. Shipboard investigations showed that the elevated $pCO_2$ only enhanced primary productivity in waters with a relatively higher availability of nutrients; in oligotrophic waters, the acidic stress associated with the elevated $pCO_2$ decreased it [84]. The involved mechanisms could be enhanced respiratory carbon loss during the night period [85]. On the ecosystem level, changes in the $CO_2$ concentration affects species composition and succession [81], since the acidic stress may affect the consumers' grazing capacity [86].

Many aquatic organisms possess an exo- or endoskeleton of calcium carbonate. Even though calcification requires high metabolic demands, it reduces grazing pressure [87].

Some Rhodophyta, such as *Jania*, *Ellisolandia* and *Corallina* [88,89], and Chlorophyta, including *Halimeda*, *Codium*, *Halicoryne* and *Acetabularia* [90], are characterized by incrustation of calcium carbonate in the cell wall. Moreover, the Phaeophyta of the genus *Padina* incorporate calcium carbonate in their thallus [91]. Ocean acidification impairs the process of calcification [71,72,92,93]. This loss in calcification can be compensated at higher metabolic costs, thus decreasing productivity and growth rate [94]. Other aquatic primary producers, such as unicellular and filamentous cyanobacteria, also utilize calcification [95], in addition to eukaryotic phytoplankton groups such as Cocolithophorides [81], e.g., *Emiliania* [71,96], which have been the major calcium carbonate producers in the ocean since the mid-Mesozoic [97]. According to these authors, field research in the deep ocean showed that the coccolith mass has increased by 40% over the past 220 years.

More recent detailed experiments using shipboard and mesocosms experiments gave divergent results: some measurements showed increased productivity and growth under enhanced $CO_2$ concentrations [98], while other researchers identified some inhibition,

which could be due to augmented respiration and photorespiration [8,99]. The observed effect depends on other environmental factors, such as nutrient availability and irradiance, and on species and phenotype [100–102].

Many zoological taxa use calcium carbonate for incrustations and accrustations, protecting against predators. A meta-analysis showed that ocean acidification decreases coral calcification, but the degree of the decrease is still uncertain [103,104]. Corals might be capable of adapting to lower pH values since samples collected from a site with naturally lower pH showed a higher degree of calcification and growth rate than samples from a site with a higher pH value [105]. Other work indicated that coral calcification is strongly reduced when the water is slightly acidified, but the inhibition is mainly brought about by $CO_3^{2-}$ and not pH, as shown by using specific buffers in lab experiments [106]. Increased nutrient availability did not affect the degree of calcium incorporation into the aragonite which makes up the skeleton of *Acropora muricata* [107]. In contrast, the zooxanthellae do not seem to be much affected by changes in the carbon chemistry [108].

Other animals which produce calcium carbonate skeletons include tube worms, bivalves, snails, cephalopods, echinoderms, crustaceans and fish [109]. Ocean acidification threatens the integrity of the tubes of the serpulid tubeworm *Hydroides elegans* [110], but warming and reduced salinity mitigate the adverse effect in this polychaete [111]. Larvae of the Pacific oyster *Crassostrea gigas* utilize the same energy expenditure under acidification condition but show a 56% reduction of the aragonite shell mass; however, large differences were found among genotypes [112]. Ocean acidification has multiple effects on bivalve larvae: in addition to effects on early and later shell development, the respiration rate is elevated under a low pH, and feeding is most sensitive to $pCO_2$ [79]. In mud snails (*Tritia obsoleta*), acidification alters the predator–prey relationships since it delays or even inhibits escape responses in the snails when attacked by mud crabs, which can significantly alter food webs at decreasing pH values [113]. Acidification has been found to impact the early development of cephalopods, e.g., by altering the accumulation of trace elements [114]. Other studies suggest few effects on cephalopods and crustaceans [115]. Many fish seem to counter acidification effects by adjusting the acid–base balance, but at a low pH, otolith growth, mitochondrial function and metabolic rate have been found to be affected [116]. In addition, changes in neurosensory and behavioral endpoints have been documented.

## 4. Ocean Deoxygenation and Its Effects

Dissolved $O_2$ (DO) levels in the oceans are decreasing due to progressive ocean warming, which is known as ocean deoxygenation. The global ocean has lost about 2% $O_2$ per decade since 1960 in terms of the total ocean inventory [60,117], and the surface ocean $O_2$ levels have been projected to drop to about 200 μmol $L^{-1}$ by the end of this century ($-5$ μmol $kg^{-1}$ per decade) [117,118]. Ocean warming decreases the $O_2$ solubility, hindering ventilation to deeper layers [119]. Consequently, the oxygen-minimum zones have spread out horizontally and vertically, along with ocean deoxygenation [118,120,121]. In addition, higher temperatures and eutrophication augment bacterial growth, which increases $O_2$ consumption by respiration [122].

Hypoxic waters (dissolved $O_2 < 63$ μM or 2 mg $L^{-1}$) occur naturally in natural waters, and global warming, as well as eutrophication, is supposed to raise their spatial extent and severeness [118,123–125]. While hypoxia has been considered exclusive to deeper waters, near-surface hypoxic waters (<20 m) are often observed in estuaries [126], coastal waters [127] and upwelling regions [128]. Lowered dissolved $O_2$ levels of waters are usually enhanced by the heterotrophic degradation of dissolved organic matter under warming influences, leading to lowered pH and elevated $CO_2$, as well [129–131]. Either from the viewpoint of ocean deoxygenation and acidification or from that of hypoxia, reduced $O_2$ availability and increased $CO_2$ (lowered pH) are co-varying factors.

It is known that reduced $O_2$ levels are harmful, and hypoxia is detrimental to most marine animals [132,133]. However, natural phytoplankton assemblages and the diatom *T. weissflogii* have been recently shown to benefit from reduced $O_2$ concentrations by en-

hancing CCMs and by increasing carbon fixation efficiency [134], though such stimulating effects were moderated by ocean acidification treatment. Nevertheless, even under the influence of ocean acidification, deoxygenation can accelerate phytoplankton photosynthesis, consequently "re-oxygenating" in illuminated waters, which thus may progressively alleviate the impacts of deoxygenation on animals. In addition, reduced $O_2$ availability has been shown to enhance the photosynthesis and $N_2$-fixation of *Trichodesmium*, a marine diazotroph without heterocysts contributing to about half of oceanic $N_2$ fixation per year (Li, He, and Gao. 2022, to be published). Under the future scenarios of ocean deoxygenation and acidification, *Trichodesmium* would decrease its mitochondrial respiration by 5% and increase its $N_2$-fixation by 49% and the $N_2$-fixation quotient by 30% with a 16% decline of $pO_2$ and 138% rise of $pCO_2$ by the end of this century. As increased temperature can increase iron use efficiency [135] and enhance mitochondrial respiration and then benefit $N_2$-fixation of *Trichodesmium*; deoxygenation, along with ocean warming, may further enhance the activity of the nitrogenase and increase $N_2$-fixation of diazotrophs, as long as ocean warming does not surpass the thermal tipping point for their growth.

Dead zones appear largely lifeless [121], but algae and phytoplankton can provide oxygen to hypoxic habitats, provided that sufficient solar radiation and nutrients are available and they are in their tolerated temperature window. In contrast, animals are dependent on the availability of dissolved oxygen in the water, with the exception of zoological taxa with photosynthetic symbionts, such as corals, sponges or tunicates [136]. However, even coral reefs are threatened by accelerating ocean deoxygenation [137]. Some species are better adapted to hypoxia than others, as shown in the tolerant sea urchin *Echinometra viridis* from the Caribbean, as compared to two other species from the Pacific coast [138]. Some micrometazoan invertebrates, such as nematodes, have been found to be metabolically active at $O_2$ concentrations below 1.8 $\mu$mol $L^{-1}$ [139]. Zooplanktonic resting stages use cytochrome c oxidase as a sensor for the oxygen concentration signaling to exit dormancy [139]. Dead zones are also found in rivers and lakes, where they threaten the development of fish eggs, such as those of the grass carp (*Ctenopharyngodon idella*) [140].

## 5. Effects of Solar UV Radiation

Short-wavelength solar radiation has many—mostly deleterious—effects on the aquatic biota. Today, radiation at wavelengths below 280 nm (UV-C) is quantitatively absorbed by oxygen and the ozone in the stratosphere and does not reach the Earth's surface, but it was a decisive component during the evolution before photosynthetically produced oxygen accumulated in the atmosphere, and, as a consequence, ozone became present in the stratosphere [141]. UV-B (280–315 nm) is partially absorbed in the atmosphere, while UV-A (315–400 nm) reaches the Earth's surface almost quantitatively [142]. The irradiance impinging on the Earth's surface is controlled by the solar zenith angle, atmospheric ozone, clouds, aerosols, surface albedo and height above sea level.

In terrestrial ecosystems, solar irradiances are controlled by circadian and annual changes and modulated by clouds and precipitation, and they are modified by global climate change. In addition, irradiances in aquatic ecosystems are modified by the tidal rhythm [143] and strongly controlled by the transparency of the water. Part of the radiation is reflected before penetrating into the water in dependence of the solar zenith angle and the smoothness of the water surface. Inside the water column, the radiation is attenuated by particulate and dissolved inorganic and organic matter. The inorganic material consists of sand and silt, especially in freshwater and coastal habitats [144]. Bacteria, phytoplankton and zooplankton form the particulate organic material (POM), while dissolved organic material is derived from decaying organisms and terrestrial runoff. Climate change affects the timing and amount of input by terrestrial runoff. UV radiation, in turn, photobleaches the dissolved organic matter (DOM) and breaks it down so that it can be more easily taken up by microorganisms. A large portion of DOM consists of chromophoric dissolved organic matter (cDOM), which controls the UV transmission in surface waters [145]. In the open ocean, cDOM and pigments are derived from the decay of marine producers such as

algae and phytoplankton [146]. The browning of surface waters in North American and European boreal lakes is due to atmospheric deposition and surface runoff [147]. UV-B radiation has been reported to decrease by 12 to 39% over the period 1961–2014 for three lakes in Eastern and Southwestern China, resulting from decreased transparency [148]. In contrast, in most continental US lakes, the transparency has increased since 1984 [149,150], which was also found in 153 large lakes in China [151,152]. Moreover, in coastal waters, extreme climate events, such as flooding, result in changes in the transparency, such as the coastal darkening observed in the North Sea [153]. cDOM is also the main factor reducing UV transparency in oligotrophic waters such as the Red Sea [154] and waters around the Great Barrier Reef [155].

In the mixed layer, the organisms are constantly moved around either by the action of wind and waves [156] or by active motility based on flagella or cilia or changes in buoyancy. Due to this scenario, the organisms are exposed to high UV radiation when close to the surface, and this can cause lesions. When they are moved to the bottom of the mixed layer, they have time to repair the damage [157]. The Montreal protocol and its amendments have succeeded in limiting the stratospheric ozone-depleting substances so that damage on the biota by solar UV radiation is limited [158] but still a stress factor.

Excessive solar UV radiation damages proteins, lipids, biomembranes and other cellular organelles [159,160]. One of the main targets for short-wavelength radiation is the DNA [161]. The most common damage is the formation of cyclobutane pyrimidine dimers (CPD) [162,163]. Cells have the capability to repair these lesions by using the enzyme photolyase, which utilizes the energy of UV-A and blue-light photons to split the dimers [164]. If not repaired the lesions may lead to mutations and death. Since the splitting of dimers is based on an enzymatic process, higher temperatures augment the repair [165]. In addition, there are many other repair mechanisms to deal with UV-induced DNA damage [166,167]. Another major target of solar UV radiation in aquatic primary producers is the photosynthetic apparatus [168]. The D1 protein that is responsible for the electron transport from the reaction center in photosystem II to the primary acceptor (pheophytin) is kinked by UV radiation [169]. This lesion is detected, and the damaged protein is degraded and subsequently replaced by a newly synthesized copy [170].

In cyanobacteria, UV radiation impairs motility and orientation [171,172] and bleaches photosynthetic pigments such as the phycobiliproteins [173]. In addition to enzymatic repair, cyanobacteria and many other phytoplankton have developed protective UV-absorbing pigments to mitigate UV-induced damage [174,175], including an array of small-molecular-weight pigments, mycosporine-like amino acids (MAAs) of which more than 20 have been isolated and characterized [176,177] (Figure 3). Animals are not capable of producing MAAs, since they lack the shikimate pathway [178]. However, many animals take up the MAAs with their diet, incorporate them in their tissues and use them for UV protection, as shown for zooplankton [179], sea urchins [180] and crustaceans [181]. Being natural products, MAAs are also being used in commercial sunscreens, replacing artificial organic absorbers [182]. In addition, only cyanobacteria produce another group of UV-screening pigments, scytonemins [183]. In addition to absorption in the UV-A and UV-B region, these substances have a strong absorption peak in the UV-C [184] which is probably a reminder of the fact that these organisms developed before there was oxygen and ozone in the atmosphere and, consequently, wavelengths <280 nm could reach the Earth's surface [185].

Organism groups, species and even cultivars vary in their sensitivity to solar UV-B radiation [186]. Generally speaking, primary producers in coastal habitats are more sensitive than those in open ocean waters, but due to the lower transparency of these waters, they are exposed to lower irradiances of short wavelength radiation [187]. Among macroalgae, red seaweeds are very sensitive, and, consequently, many species are found at a greater depth [188], while many green macroalgae are more tolerant and are found higher up in the eulitoral and sometimes even above the water during low tide [7]. The

differences in UV sensitivity between cultivars of the same species can be explained by adaptation, nutrient supply and differences in experimental setup.

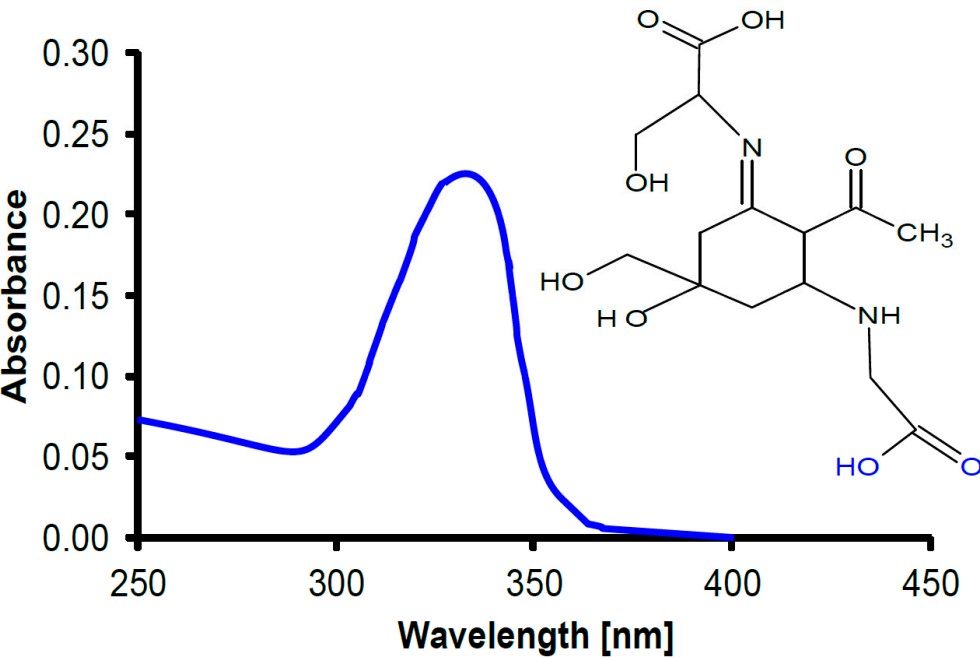

**Figure 3.** Chemical structure and absorption spectrum of the mycosporine-like amino acid shinorine found in many prokaryotic and eukaryotic phytoplankton species, as well as some macroalgae.

## 6. Effects of Multiple Drivers

Solar UV radiation (UVR, 280–400 nm) can damage DNA [159,160] and repress its repair in phytoplankton [189]. While UV-B irradiance represents less than 1% of the total solar energy, it is commonly more harmful than UV-A, as UV-B photons are more energetic than those of UV-A, which is about 6–8% of the total solar energy in subtropical areas. In addition, UVR can also generate active free oxygen radicals that lead to oxidative stress [159], lowering photosynthetic rates [190]. However, UVR, especially UV-A, may enhance photosynthesis of phytoplankton assemblages [191] and macroalgae [192,193]. Through historical adaptation, phytoplankton and macroalgae are able to cope with UVR, mainly by synthesizing UV-screening pigments such as MAAs and by eliminating active oxygen free radicals and repairing damaged proteins and DNA [159]. Nevertheless, in the surface layer of the water column, UVR can decrease photosynthetic carbon fixation by up to 30% [157].

Ocean warming and acidification expose phytoplankton cells to higher temperatures and lower pH. The diatom *Skeletonema costatum* has been reported to increase the activity of periplasmic carbonic anhydrase (CAe) when exposed to moderate UVR levels and to raise its CCMs efficiency [84,194]. For the red tide alga *Phaeocystis globosa* grown under OA at 1000 µatm pCO₂ and full spectrum solar radiation with UVR, its photosynthetic efficiency showed the lowest values at noon [195]. Under fluctuating solar radiation, OA acted antagonistically with mixing to enhance photosynthetic carbon fixation by UV-A, while moderating the photochemical inhibition caused by UV-B in a coccolithorphorid [196], suggesting that the combination of OA and UVR results in differential effects for different UV wavelengths.

The effects of OA and UV synergistically decrease the calcification of algal calcifiers [71]. OA decreases calcification in coralline algae and coccolithophores and consequently makes their calcified layers thinner, exacerbating the harmfulness of UVR [197,198]. The combined effects of OA and warming, PAR light intensity fluctuation and UVR on a coccolithophorid *Gephyrocapsa oceanica* have been examined by Jin et al. [199], showing that the combination of OA and warming synergistically increased the photochemical

performance and photosynthesis under a lower frequency of light fluctuations, with less impacts of UVR being observed.

Warming is suggested to alleviate UV-related damage due to increased activities of enzymes involved in repair processes [200,201]. When the diatom *Phaeodactylum tricornutum* had acclimated to two $CO_2$ concentrations (390 and 1000 µatm) for more than 20 generations, OA treatment enhanced the non-photochemical quenching (NPQ) of the cells and partially counteracted the damage of UVR to PSII, which, however, was moderated by warming treatment [202]. Contrastingly, OA treatment with elevated $pCO_2$ of 700 µatm interacted with both incident UV-A and UV-B levels to decrease the photosynthetic carbon fixation of coastal phytoplankton assemblages in the South China Sea [203].

Many microalgae and the spores of many macroalgae migrate or move vertically and horizontally by flagellar movement or due to water motions. They are supposed to swim toward light via phototaxis under low light intensities but away from high light intensities and UVR in a similar way to how copepods do (see [204] and the literature therein). The swimming ability of flagellated microalgae was shown to be impaired by OA treatment, suggesting that their capacity to survive will be endangered in future high $CO_2$ oceans [205], especially when they are simultaneously exposed to top-down (associated with warming and UV radiation) and bottom-up (due to low levels of pH and $O_2$ in subsurface hypoxic waters) pressures.

Aquatic consumers are also threatened by biotic and abiotic factors, including increasing temperatures, acidification, low nutrient availability and excessive UV radiation, which all affect their survival, behavior, reproduction and population dynamics [206]. Zooplankton is not only affected by changing temperatures and UV radiation. Warmer waters increase the toxicity of lithium and mixtures of microplastics and lithium [207]. Lithium concentrations ≤0.1 mg/L were found to kill populations of *Daphnia magna*. High irradiances (but low UV) had the same synergistic effect. Hypolimnetic anoxia affects diel circadian vertical migrations and thus critically interferes with their grazing capability, affecting the food web [208]. Many other environmental factors are changed by increasing temperatures, which, in turn, alter the behavior, survival and growth of both freshwater and marine zooplankton by, for example, the exposure to damaging UV radiation and predators [209].

## 7. Conclusions and Perspectives

It is mandatory not only to analyze the effects of individual environmental factors on species composition, community assemblages and the productivity of aquatic organisms, but also to study the interactions of these factors, such as water temperature, ocean acidification, solar UV radiation, nutrient availability and deoxygenation in their habitat under natural conditions. Aquatic organisms have developed many mechanisms to mitigate the negative effects of environmental factors such as poleward migration of both sessile and motile species to escape extreme temperatures, rapid repair mechanisms and UV-absorbing pigments. Anthropogenic global climate change is an important factor which changes the conditions for growth, reproduction and productivity in an unprecedented fast manner. Predictions for the near future include changes in species composition and sequential development of primary producers with severe consequences for the food chain and food availability for predators, including humans. However, many organisms have developed mitigating strategies to escape exceeding the temperature by poleward migration, exposure to extreme solar visible and UV radiation by efficient repair mechanisms and the production of UV-absorbing pigments, as well as ocean acidification due to increasing $CO_2$ concentration in the top water layer. In addition, genetic adaptation over hundreds or thousands of generations is an effective means to increase the tolerance toward extreme environmental factors which are increasingly worsened by the effects of anthropogenically produced climate change.

In order to generate ecologically significant results and to provide meaningful predictions for the development of aquatic ecosystems under the effects of multiple stressors

related to climate change conditions, future research has to focus on experiments and analyses under natural conditions in aquatic habitats. This is not an easy task given the vast spaces of the oceans and the comparatively low concentrations of primary producers and their predators. However, this may be achieved by using sophisticated technologies and modern equipment such as molecular genetics and sensitive equipment. Even though there have been increases in knowledge in understanding ecological effects of multiple stressors, the efforts made so far to apply the acquired knowledge to designing management actions on aquatic ecosystems have not yet been applied to remediating climate changes, since oceanic sinks and sources of the greenhouse gases on a global scale are still lacking, considering recent findings that phytoplankton release methane as a byproduct during photosynthesis [210,211]. In addition, regional responses to climate-change drivers differ due to biological, chemical and physical conditions that change spatiotemporally; therefore, integrative analyses with modeling predictions are expected in future studies to look into the impacts of climate-change drivers on different waters and/or different types of ecosystems.

**Author Contributions:** Both authors equally contributed to the conceptualization and writing. All authors have read and agreed to the published version of the manuscript.

**Funding:** This research received no external funding.

**Data Availability Statement:** Not applicable.

**Acknowledgments:** This study was supported by the national key R&D program (2022YFC3105303) and National Natural Science Foundation of China (41721005, 41720104005, 41890803). The authors are grateful to the laboratory engineers Xianglan Zeng and Wenyan Zhao for their logistical and technical support.

**Conflicts of Interest:** The authors declare no conflict of interest.

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
