# Peer review of "Aquatic Productivity under Multiple Stressors"

_water, doi:10.3390/w15040817_

Round 1
Reviewer 1 Report
Dear Authors:
Overall, the topic is interesting. I found some of the description of the paper to be not that detailed, while the description and explanation of some of the very important points were missing or lacking. Review comments must be addressed first to accept this manuscript for publication. Also, the novel point in this study should be emphasized. Why is this study important and how is this research different from other previously reported ones.
1) Even though there is an increase in documented research on multiple stressor effects, the efforts made so far to apply these acquired knowledge on designing management actions on aquatic ecosystem have resulted in small improvements and below the target expectations. The author(s) should also add in the revised manuscript the current management strategies that can be done to lessen the effects of these multiple stressors in the aquatic ecosystem.
2) The different multiple stressors stated in this manuscript are known to act simultaneously at varying time and spatial scales, with their effects being susceptible to differ with local natural conditions, climate changes, and spatial scale. How does the author addressed these concerns in the manuscript. We need more explanation on how to address this and would like to see I in the revised manuscript.
3) The author(s) focused more on enumerating the multiple stressors. I suggest that they should include the challenges in implementing efficient management practices, such as adapting monitoring strategies to new evidence on the relationships between ecological responses and multiple-stressor interactions, improving our knowledge and understanding of the mechanisms involved in stressor interactions, and shifting the focus from ecosystem degradation pathways (which recently being considered as a primary focus of research done on multiple stressors) to the processes that govern recovery.
4) I suggest that the author(s) make a table that will summarize the effects of the different multiple stressors of the aquatic ecosystem including previous studies done on similar topic.
5) The manuscript provided a representative view of the effects of multiple stressors in aquatic ecosystem. It is also important to include how these multiple stressors are currently being addressed by researchers, managers and decision-makers. The authors should discuss the current scenario on how international cooperation between local stakeholders and researchers of different countries with natural resource limitations might help to contribute in environmental sustainability of their aquatic ecosystem.
Author Response
Overall, the topic is interesting. I found some of the description of the paper to be not that detailed, while the description and explanation of some of the very important points were missing or lacking. Review comments must be addressed first to accept this manuscript for publication. Also, the novel point in this study should be emphasized. Why is this study important and how is this research different from other previously reported ones.
1) Even though there is an increase in documented research on multiple stressor effects, the efforts made so far to apply these acquired knowledge on designing management actions on aquatic ecosystem have resulted in small improvements and below the target expectations. The author(s) should also add in the revised manuscript the current management strategies that can be done to lessen the effects of these multiple stressors in the aquatic ecosystem.
Response: we acknowledged this by adding the following at the end of perspective part: “Even though there have been increased knowledges in understanding ecological effects of multiple stressors, the efforts made so far to apply these acquired knowledges on designing management actions on aquatic ecosystems have not yet applied in remediating climate changes, since oceanic sinks and sources of the greenhouse gases on a global scale is still lacking, considering recent findings that phytoplankton release methane as a byproduct during photosynthesis [210,211].”we added the additiona references,that is [210,211]
210 Günthel M, Klawonn I, Woodhouse J, Bižić M, Ionescu D, Ganzert L, Kümmel S, Nijenhuis I, Zoccarato L, Grossart HP, Tang KW. 2020. Photosynthesis-driven methane production in oxic lake water as an important contributor to methane emission. Limnology and Oceanography 65: 2853-2865.
211 Bižic M, Klintzsch T, Ionescu D, Hindiyeh M Y, Günthel M, Muro-Pastor AM, Eckert W, Urich T, Keppler F, Grossart HP. 2020. Aquatic and terrestrial cyanobacteria produce methane. Science Advances 6: eaax5343.
2) The different multiple stressors stated in this manuscript are known to act simultaneously at varying time and spatial scales, with their effects being susceptible to differ with local natural conditions, climate changes, and spatial scale. How does the author addressed these concerns in the manuscript. We need more explanation on how to address this and would like to see I in the revised manuscript.
Response: it is true that regional responses to climate change drivers differ due to biological, chemical and physical conditions that change spatio-temperally. In response to this, we added at the end of the paper as follows: “In addition, regional responses to climate change drivers differ due to biological, chemical and physical conditions that change spatio-temporally, therefore, integrative analysis with modeling predictions are expected in future studies to look into the impacts of climate change drivers on different waters and/or different types of ecosystems”
3) The author(s) focused more on enumerating the multiple stressors. I suggest that they should include the challenges in implementing efficient management practices, such as adapting monitoring strategies to new evidence on the relationships between ecological responses and multiple-stressor interactions, improving our knowledge and understanding of the mechanisms involved in stressor interactions, and shifting the focus from ecosystem degradation pathways (which recently being considered as a primary focus of research done on multiple stressors) to the processes that govern recovery.
Response: agree. This constructive comments would be useful for future modeling scientists to integratively analyze the data obtained by monitoring and via experiments. For this review, what we can do is to provide basic understanding of effects and involved mechanisms.
4) I suggest that the author(s) make a table that will summarize the effects of the different multiple stressors of the aquatic ecosystem including previous studies done on similar topic.
Response: we agree that a table would be useful. Considering the large span of biodiversity and tremendous documented literatures, it is almost impossible to provide a reliable table to reflect different bio-groups and different factor even combinations of different factors. Therefore, we decided not to, since our primary expertise is limited to primary producers, and because tables for different groups can be found the reviews cited.
5) The manuscript provided a representative view of the effects of multiple stressors in aquatic ecosystem. It is also important to include how these multiple stressors are currently being addressed by researchers, managers and decision-makers. The authors should discuss the current scenario on how international cooperation between local stakeholders and researchers of different countries with natural resource limitations might help to contribute in environmental sustainability of their aquatic ecosystem.
Response: this comment is highly evaluable. However, as responded above, to find how these multiple stressors are currently being addressed by managers and decision-makers is out of our scope of the review, and should be done with help of social statistics. How different studies lead to controversial findings were discussed in our review.

Reviewer 2 Report
Major comments:
1. Figure 1 is not cited in the text.
2. Figure 3 is cited in the text, but no Figure 3 can be found in the text.
Minor editorial comments:
1. lines 10, 36, 52, 61, 68, 92, 163, 175, 180, 214, 237,: delete space between value and %
2. line 83: add a comma after 1979
3. line 140: spelling of El Nino should be fixed
4. line 170: IPCC AIF1 ?
5. line 199: add comma after level
6. line 213: add omma after authors
7. line 218: comma should be period after [8, 99]
8. line 231: add space after chemistry
9. line 241: add comma befoer acidification
10. line 278: (Li, He, Gao, 2022, to be published): Is this journal allowed to cite unpublished manuscript?
11. line 301: add comma after Today
12. line 323: add comma after ocean
13. line 328: add comma after lakes
14. line 329: add comma after waters
15. line 334: add comma after layer
16. line 336: add comma after scenario
17. line 366: add comma after region
18. line 371: add comma after speaking
19. line 373: add comma after macroalgae
20. lines 388-389: deleet (mycosporine-like amino acids)
21. line 431: foound should be found?
22. line 477: global climate change on cyanobacteria
23. line 481: In of Sea Salt?
24. line 494: Prochlorococcus and Synechococcus PLoS ONE
25. line 493: pelagic marine organism
26. line 496: Pak I Biol
27. lines 543, 583, 586: Is this a book? if so, add publisher or add source of this information
28. line 592: Monit Marine Pollut
29. line 594: Anthrop Pollut Aquat Ecosyt
30. line 629: PLoS ONE
31. line 634: Ann Rev Plant Biol
32. lines 649-652: scientific names should be italic
33. line 686: Coral Reefs: An Ecosystem in Transition
34. line 694: PLoS One
35. lines 700, 703: scientific names should be italic
35: line 714: add USA after Sci.
36. line 717: An Environmental Evaluation
37. line 739: Global Change Biol
38. line 741: Commun Biol
39. line 774: add space after Sci
40. line 792: add space after Ecol
41. line 833: Cyanobacteria
42. line 844: add space after Aquat
43. lines 848-849: Cyanobacterial Biology
44. line 871: title should not be large capial, scientific name should be italic
45. line 872: Photobiol
46. lines 878, 882: scientific names should be italic
47. line 892: PLoS ONE
48. line 894: scientific name should be italic
49. line 902: title should not be large capital
Author Response
The authors thank the reviewer for the detailed and helpful comments.
1. Figure 1 is not cited in the text. Now cited
2. Figure 3 is cited in the text, but no Figure 3 can be found in the text. I will submit Fig. 3 again.
Minor editorial comments:
1. lines 10, 36, 52, 61, 68, 92, 163, 175, 180, 214, 237,: delete space between value and % Done
2. line 83: add a comma after 1979 Some English grammars do not want a comma in this context, but I have inserted one
3. line 140: spelling of El Nino should be fixed Done
4. line 170: IPCC AIF1 ? The IPCC uses several models to predict the severeness of climate change. The A1F1 assumes that the emissions will continue as before
5. line 199: add comma after level done
6. line 213: add omma after authors done
7. line 218: comma should be period after [8, 99] done
8. line 231: add space after chemistry done
9. line 241: add comma befoer acidification done
10. line 278: (Li, He, Gao, 2022, to be published): Is this journal allowed to cite unpublished manuscript? will check with the journal
11. line 301: add comma after Today done
12. line 323: add comma after ocean done
13. line 328: add comma after lakes done
14. line 329: add comma after waters done
15. line 334: add comma after layer done
16. line 336: add comma after scenario done
17. line 366: add comma after region done
18. line 371: add comma after speaking done
19. line 373: add comma after macroalgae done
20. lines 388-389: deleet (mycosporine-like amino acids) done
21. line 431: foound should be found? done
22. line 477: global climate change on cyanobacteria done
23. line 481: In of Sea Salt? done
24. line 494: Prochlorococcus and Synechococcus PLoS ONE done
25. line 493: pelagic marine organism done
26. line 496: Pak I Biol done
27. lines 543, 583, 586: Is this a book? if so, add publisher or add source of this information done
28. line 592: Monit Marine Pollut done
29. line 594: Anthrop Pollut Aquat Ecosyt done
30. line 629: PLoS ONE done
31. line 634: Ann Rev Plant Biol done
32. lines 649-652: scientific names should be italic Anthrop Pollut Aquat Ecosyt done
33. line 686: Coral Reefs: An Ecosystem in Transition done
34. line 694: PLoS One done
35. lines 700, 703: scientific names should be italic done
35: line 714: add USA after Sci. done
36. line 717: An Environmental Evaluation done
37. line 739: Global Change Biol done
38. line 741: Commun Biol done
39. line 774: add space after Sci done
40. line 792: add space after Ecol done
41. line 833: Cyanobacteria done
42. line 844: add space after Aquat done
43. lines 848-849: Cyanobacterial Biology done
44. line 871: title should not be large capial, scientific name should be italic done
45. line 872: Photobiol done
46. lines 878, 882: scientific names should be italic done
47. line 892: PLoS ONE done
48. line 894: scientific name should be italic done
49. line 902: title should not be large capital done
Round 2
Reviewer 1 Report
The authors addressed the comments and suggestions of the reviewer in the revised manuscript.